# Easy Synthesis of Doped Graphitic Carbon Nitride Nanosheets as New Material for Enhanced DNA Extraction from Vegetal Tissues Using a Simple and Fast Protocol

Manuel Eduardo Martínez-Cartagena [1] , Juan Bernal-Martínez [2,†], Arnulfo Banda-Villanueva [3],
Víctor D. Lechuga-Islas [3] , Teresa Córdova [3] , Ilse Magaña [3] , José Román Torres-Lubián [3] ,
Salvador Fernández-Tavizón [3], Jorge Romero-García [3,†], Ana Margarita Rodríguez-Hernández [3,*] and
Ramón Díaz-de-León [3,*]

1   Department of Engineering and Materials Chemistry, Centro de Investigación en Materiales Avanzados S.C.
    CIMAV, Miguel de Cervantes Saavedra, No. 120, Chihuahua 31136, Mexico
2   Laboratory in Biomedicine and Nanotechnology, Cañada Honda, No. 129, Ojo Caliente,
    Aguascalientes 20196, Mexico
3   Research Center in Applied Chemistry (CIQA), Enrique Reyna Hermosillo, No. 140, Col. San José de los
    Cerritos, Saltillo 25294, Mexico
*   Correspondence: ana.rodriguez@ciqa.edu.mx (A.M.R.-H.); ramon.diazdeleon@ciqa.edu.mx (R.D.-d.-L.)
†   These authors contributed equally to this work.

**Abstract:** Conventional and commercially available DNA extraction methods have several limitations regarding, for instance, contamination, and complex and slow precipitation and recovery processes. Herein, we report the synthesis of oxygen and phosphorus-doped Graphitic carbon nitride structures (g-POCN), via a novel Zinc-catalyzed one-pot solvothermal approach, and its application in the extraction of genomic DNA (gDNA) from a vegetal matrix (*P. argentatum*). Experimental and molecular modeling analyses demonstrate the high affinity of gDNA with g-POCN, which provided highly efficient gDNA extraction processes, with extraction yield, as well as integrity and quality of the extracted gDNA, comparable or superior to a commercial extraction kit and isopropanol extraction. Moreover, under suitable elution conditions, this method allows the easy removal of high concentrations of gDNA from g-POCN, rendering this method as a low-cost, simple, and fast approach for the extraction of even small amounts of gDNA. Remarkably, the extracted gDNA shows no degradation, and no inhibition of the polymerase chain reaction. Therefore, g-POCN represents a promising material for the highly efficient, cost-effective, and biocompatible extraction of DNA, which could stimulate research focused on broad DNA sources, e.g., RNA extraction, plasmids, ssDNA, etc.

**Keywords:** DNA extraction; graphitic carbon nitride; DNA purification; PCR; solvothermal synthesis

## 1. Introduction

Graphitic carbon nitride (typically denoted as g-C$_3$N$_4$ or g-CN) has become an important material in nanotechnology development. The applications of graphitic carbon nitride have been mainly focused on the field of photocatalysis [1]. This technique, which is used to produce energy, is a cleaner and more efficient alternative to traditional energy sources. The typical route to obtain g-CN is by the pyrolytic synthesis, in detriment. However, this method occurs at a high temperature, requires a controlled atmosphere, and the doping process is difficult to achieve [2]. In recent years, several reports have shown new synthesis pathways to obtain g-CN, including solvothermal synthesis, thermal shrinkage polymerization, solid phase synthesis, gas phase synthesis, and electrochemical deposition [3]. Solvothermal synthesis involves the reaction or transformation of one or more precursors in the presence of a solvent at a temperature higher than its boiling point in a

sealed container [4]. This method is a promising way to synthesize doped g-CN derivatives because it requires a lower temperature, thus demonstrating low energy consumption and offering the possibility to reuse the solvents [5]. Additionally, this method has shown a higher yield than other methods [6].

Doped g-CN offers a wide variety of tunable properties with multifunctional applications in wastewater and environmental treatment, solar energy [7], catalysis, imaging, white-light-emitting diodes [8], and emerging biomedical applications such as biosensors, biotherapy, and DNA extraction [7]. In this context, DNA extraction represents the limiting step in any genetic lab work. This process is crucial because it affects the purity and integrity of the extracted materials and the subsequent steps of DNA amplification, sequencing, or gene manipulation [9]. Nucleic acids' extraction is a process that begins with a sample, usually in the form of cells, tissue, or blood. The extraction starts by breaking the cell and releasing its contents. Next, enzymes are used to break down the proteins and fats of the cell into smaller molecules. Finally, these small molecules are separated from each other using solvent methods or a solid matrix filtration kit [10].

However, DNA extraction has some drawbacks: First, the DNA can be contaminated by other substances that are not purified from the sample. Secondly, there is a risk of losing DNA due to precipitation or during the recovery process. Thirdly, there is a risk of contamination in the lab during any step of the purification process because avoiding contamination in a laboratory is a challenging task [11]. Additionally, the extraction protocol used in the DNA isolation could represent an environmental and health risk as well as a potential contamination source during the DNA extraction due to the chemicals involved through the purification, for instance, the EtBr (ethidium bromide)–CsCl extraction method requires harmful and expensive chemicals [12]. Alkaline extraction employs NaOH and detergent; this method is typical for plasmid extraction, but it could contaminate the extraction with DNA fragments [13]. Cetyltrimethylammonium Bromide (CTAB) Extraction is a common method employed for DNA purification from plants and bacteria; however, this method requires toxic organic solvents and chemicals [14]. The phenol-chloroform method is a gold standard in DNA extraction; nevertheless, this protocol involves harmful organic solvents, which show a relevant risk to health and the environment [14]. Recently DNA purification pathways have been proposed based on solid matrices as functionalized cellulose, silica, polystyrene, and magnetic beds [15–17], which represent a more friendly and harmless approach, however, the cost could be punitive.

One way to overcome these limitations, which has recently been studied, is through the use of nanomaterials. Nanomaterials are capable of interacting with DNA and enabling higher extraction yields with improved integrity. In this regard, Hashemi et al. [9] applied exfoliated graphene oxide nanoplatelets and hydrazine-reduced graphene oxide nanoplatelets to extract a bacterial plasmid. The experiment was successful because these materials are able to bind with nucleic acids. In the same context, Pham et al. [18] proposed a magnetic material for facile RNA extraction, which consisted of silica-coated magnetic beads conjugated with graphene oxide (GO). Several reports have developed DNA biosensors using graphene GO [19–22]; likewise, g-CN sheets have also been successfully employed for the fabrication of biosensors or nano-assemblies of single stranded DNA (ssDNA) [23,24]. The chemical and nano-metrical features of g-CN sheets enabled the development of a protonated g-CN-based ratiometric fluorescence platform conjugated with ssDNA (ss-DNA/g-CN). This method is based on the principle that ssDNA can be used as a universal probe to detect specific targets in various environments, such as heavy metal ions ($Hg^{2+}$) and biomolecules (Aflatoxin B1), and adenosine triphosphate (ATP) [24]. In this regard, although double-stranded DNA (dsDNA) is a more accessible and stable structure than ssDNA, due the structure, conformational flexibility and its complex association with proteins [25], there are no reports about using g-CNs for the extraction of dsDNA materials.

Herein, we report the successful synthesis of doped graphitic carbon nitride (g-POCN) structures through a one-pot solvothermal approach and using urea and phosphorus pen-

toxide as the carbon, nitrogen, and phosphorus sources. To the best of our knowledge, this process represents the first attempt at using zinc powder as a catalyst for the simultaneous oxygen and phosphorus doping of derived-carbon nitride products. The obtained g-POCN was successfully employed in the DNA extraction of vegetal matrix; a procedure that commonly shows drawbacks related to the extraction yield and minimal deleterious interaction between g-CN sheets and DNA. We conclude that g-POCN represents a promising material for the highly efficient, cost-effective, harmless, ecofriendly, and biocompatible extraction of DNA, which could be applied in broad DNA sources.

## 2. Materials and Methods

### 2.1. Materials

Urea (98%), phosphorous pentoxide (98%), toluene (99.5%), and ethanol (95%) were purchased from Sigma-Aldrich (St. Louis, MO, USA). Powdered zinc (99.995%) was provided by Quimica Monterrey (Monterrey, N.L., Mexico). All chemicals were used as received.

### 2.2. Solvothermal Synthesis of Phosphorous-Doped g-POCN Catalyzed with Metallic Zinc

In a typical experiment, 400 mg of urea, 100 mg of phosphorous pentoxide ($P_2O_5$), and 90 mg of powdered zinc were vigorously mixed (assisted by a Vortex 2 Genie (Scientific Industries, Bohemia, New York, NY, USA) at full power) with 30 mL of toluene in a glass vial for 15 min. Then, the suspension was transferred to a Teflon cylinder, sealed in a stainless-steel autoclave, and heated to 230 °C in an electric oven. After the reaction time elapsed, the resulting solid was dispersed in an aqueous solution (pH 1). The mixture was treated with an ultrasonic bath for 1 h and filtered (0.25 μm Teflon filter). The filtered solid was washed with ethanol, water, and acetone 4 times each. Finally, the filtered solid was dried for 24 h at 70 °C and recovered as a white solid named as g-POCN. Figure 1 shows a flow diagram of the general procedure followed in the investigation from the synthesis stage as well as the respective characterization and analysis.

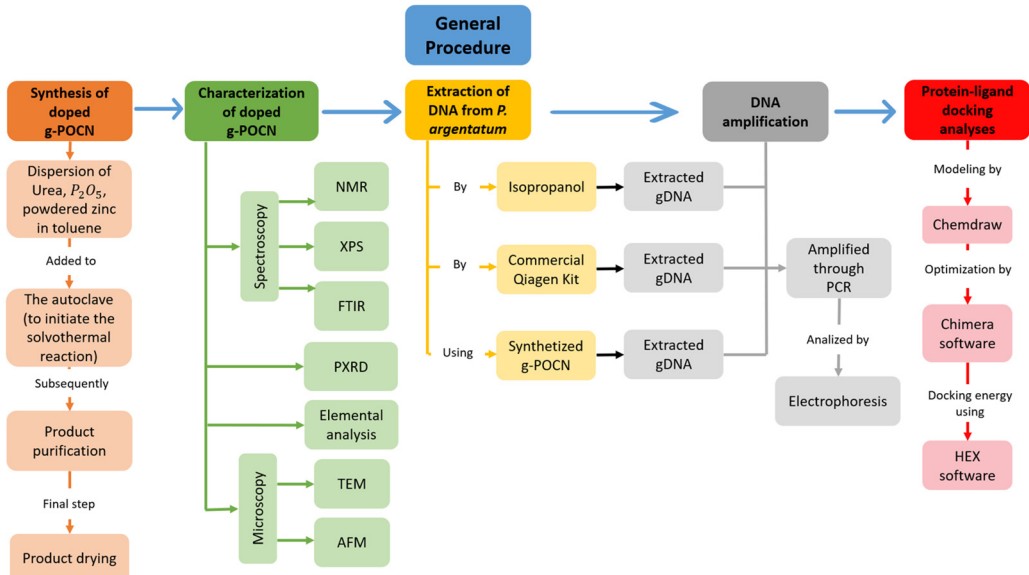

**Figure 1.** General procedure followed in the research project.

### 2.3. Nuclear Magnetic Resonance Spectroscopy (NMR)

NMR spectra were obtained on a Brucker Advance III HD Ascend 400 MHz spectrometer (Billerica, MA, USA) with a 5 mm multinuclear cryogenic probe and Z-gradient. $^1$H spectra were acquired with 30° pulses and 1-sec delay times. $^{13}$C and $^{31}$P spectra were acquired in decoupled mode with 30° pulses and 2-sec delay times. $^{15}$N spectra were

acquired using the sequence inverse gate decoupling with 90° pulse, sw = 230 ppm, TD = 3, 768, NS = 3,200, and 10 sec of delay time. In the case of the $^1$H and $^{13}$C NMR spectra, the chemical shifts were referenced with respect to the signal of the remaining non-deuterated solvent. Chemical shifts in the $^{31}$P spectra were externally referenced to $H_3PO_4$ (NMR reference standard, 85% in $D_2O$ (99.9 atom % D), NMR tube size 5 mm × 8 in, WGS-5BL coaxial NMR tube) and those of $^{15}$N were externally referenced to $CD_3NO_2$ (Nitromethane-D3, NMR, 99 atom % D, Thermo Scientific, Waltham, MA, USA). The solvents used were $D_2SO_4$ (Sigma-Aldrich, St. Louis, MO, USA, 96–98 wt.% in $D_2O$, 99.5 atom % D) and $H_2SO_4$ (Sigma-Aldrich, St. Louis, MO, USA, 99.99%), in which case a capillary containing deuterated toluene (Sigma-Aldrich, St. Louis, MO, USA, 99.6 atom % D) was used to lock and adjust the shimming of the equipment.

### 2.4. Sample Preparation

A total of 50 mg of g-POCN was subjected to an ultrasonic bath in the presence of $D_2SO_4$ until reaching the translucency of the medium. Such suspensions were highly stable over weeks without observing any precipitation and showing chemical stability after the continuous spectra acquired after one month. In cases of $P_2O_5$ samples, $D_2O$ (Sigma-Aldrich, St. Louis, MO, USA, 99.9 atom % D) was used as a solvent.

### 2.5. Elemental Analysis

For this analysis, the samples were sent to Galbraith Laboratories, who employ analytical methods for the determination of CHONP, based on the use of elemental analyzer equipment, which have international standard certification. For further information about the techniques employed, see the Supplementary Materials (Figure S1 (Supplementary Materials)).

### 2.6. Powder X-ray Diffraction (PRXD)

PXRD patterns were acquired on a Brucker D8 Advance Diffractometer (Billerica, MA, USA) with a Cu Kα radiation source (λ = 1.5418 Å). The powder samples were placed in a standard sample holder, and the measurements were made with an interval of 0.02° at a scanning speed of 10° min$^{-1}$ from 2θ = 2° to 82°.

### 2.7. Fourier Transform Infrared Spectroscopy (FTIR)

FTIR spectra were acquired using a Fischer Scientific Thermo FTIR Spectrophotometer (Watertown, MA, USA) in the attenuated total reflectance (ATR) mode, using a diamond crystal. The spectra were acquired taking an average of 32 scans with a resolution of 4 cm$^{-1}$ in a range from 400 cm$^{-1}$ to 4000 cm$^{-1}$.

### 2.8. X-ray Photoelectron Spectroscopy (XPS)

XPS measurements were performed in a Phi5000 Versa Probell equipment, ULVAC-Phi, Inc. (Kanagawa, Japan). The analysis chamber was operated at 10$^{-9}$ Torr, and the X-ray source has an aluminum anode with an energy of 1486.6 eV. Charge compensation on the surface of the samples was performed using an electron gun and an argon ion gun. To ensure that the samples were free of moisture, they were dried in a vacuum oven (0 bar for 24 h at 100 °C) before analysis.

General XPS spectra were used to determine the percentages of elements present in the samples. High-resolution XPS spectra were analyzed using a Shirley correction to subtract the background (collection of inelastic events). The deconvolution was performed using a combination of Gaussian functions keeping FWHM fixed and leaving the energy value free, with the aim of determining the bonds present.

### 2.9. Transmission Electron Microscopy (TEM)

TEM images were taken using a Titan Fei Thermo Fisher instrument (Waltham, MA, USA) with an accelerating voltage of 200 kV.

### *2.10. Atomic Force Microscopy (AFM)*

AFM micrographs were obtained using a Digital Instrument Dimension 3100 (VEECO, Santa Barbara, CA, USA). Images were obtained in the tapping mode using silicon nitride tips with a radius of curvature of 10 nm.

### *2.11. Protocol of DNA Extraction by Isopropanol*

First, plant tissue was macerated in liquid nitrogen. Next, a sample of $80 \pm 2$ mg was taken, put into a microtube, and mixed with 700 μL of preheated extraction buffer (65 °C) 0.1 M TrisHCl (Sigma-Aldrich, St. Louis, MO, USA, molecular biology grade), pH 7.5; 0.05 M EDTA (BioRad, Hercules, CA, USA) molecular biology grade), pH 8.0, 1.25% (*w/v*) SDS (Sigma-Aldrich, St. Louis, MO, USA molecular biology grade), 10 μL of β-mercaptoethanol (0.2 M, Sigma-Aldrich, St. Louis, MO, USA, 99%), and 4 μL of RNase (ROCHE, Basel, Switzerland molecular biology grade, 10 mg/mL). The mixture was homogenized in a vortex and incubated for 30 min at 65 °C. Then, 400 μL of 6M Ammonium Acetate (BioRad, Hercules, CA, USA, molecular biology grade) was added to the mixture. The resulting mixture was incubated in an ice bath for 15 min and centrifuged at 2320 g for 30 min at 4 °C. The supernatant was recovered, mixed with 650 μL of isopropanol (Sigma-Aldrich, St. Louis, MO, USA 99.5%), and incubated at 4 °C for 24 h. Next, the mixture was centrifuged at 15,000 g for 30 min at 4 °C. The formed DNA pellet was washed twice with 500 μL of 75% ethanol (Sigma-Aldrich, St. Louis, MO, USA molecular biology grade) and centrifuged at 12,000 g for 1 min. After that, the supernatant was discarded, and the pellet was allowed to dry carefully. Finally, the DNA was eluted by adding 100 μL of sterile milliQ $H_2O$. The extracted DNA was stored at $-20$ °C. The quality and quantity of extracted DNA were determined spectrophotometrically at 280 and 260 nm (Synergy Microplate Spectrophotometer, Take3 micro-volume plate; BioTek, Winooski, VT, USA).

### *2.12. DNA Extraction with g-POCN*

A similar extraction protocol as described above was followed. In this case, the recovery of the cell lysis supernatant was modified: after centrifugation at 2320 g for 30 min at 4 °C, the recovered supernatant was mixed with 30 or 50 μL of g-POCN (250 mg mL$^{-1}$) and incubated for 30 min with shaking at 4 °C. After centrifugation at 12,000 g for 15 min at 4 °C, the supernatant was discarded. The obtained pellet was washed 2 times with 500 μL of 75% ethanol and centrifuged at 1200 g for 1 min at 4 °C. Then, 150 μL of Elution Buffer TE (10 mM Tris-HCl pH 8.8 + 1mM EDTA) was added to the recovered and dried pellet. The microtube was incubated for 10 min with shaking at 65 °C. After that, it was centrifuged at 12,000 g for 3 min at 4 °C, and the supernatant was carefully recovered. Note: it is possible to obtain a second DNA elution from the pellet by repeating the protocol from the addition of the elution buffer to the g-POCN pellet discarded in the last step of the protocol. The quality and quantity of extracted DNA were determined spectrophotometrically at 280 and 260 nm (Synergy Microplate Spectrophotometer, Take3 micro-volume plate; BioTek, Winooski, VT, USA), and the DNA was analyzed by electrophoresis onto a 1% agarose gel in TAE (Tris-acetate EDTA; Thermo Fisher Scientific, Waltham, MA, USA) buffer; the gel was run at 100 V for 60 min.

### *2.13. DNA Extraction Protocol with Qiagen Kit*

In the same way as described above, plant tissue was macerated in liquid nitrogen. Next, a sample of $80 \pm 2$ mg was taken, put into a microtube to perform DNA extraction by using a Qiagen DNeasy Plant Mini Kit (Qiagen N.V., Venlo, The Netherlands) following the instructions of the manufacturer.

### *2.14. DNA Amplification by PCR*

The integrity, consistency, and non-interference complex of the DNA extracted by the various methods was studied through PCR amplification.

We used the primers ITS-u1-F GGAAGKARAAGTCGTAACAAGG and ITS-u4-R RGTTTCTTTTCCTCCGCTTA to amplify a fragment of the ITS1-ITS2 region in *P. argentatum.* These primers were reported by Cheng et al. [26] generating a fragment of approximately 700 bp (0.7 kb). The amplification reaction was carried out in a 25 μL volume, using a Taq polymerase kit (Axygen, Union City, CA, USA) containing: 18.125 μL ultrapure $H_2O$, 2.5 μL PCR Buffer (50 mM Tris-HCl, 100 mM NaCl, pH ~8.3), 0.75 μL $MgCl_2$ (25 mM), 0.5 μL dNTPs mix (10 mM), 0.125 μL Taq DNA polymerase, 1 μL extracted DNA, 1 μL of each primer (10 mM). The PCR reaction–heating cycle program was developed as follows: initial denaturation 94 °C for 4 min, followed by 35 cycles at 94 °C for 30 s (denaturation), 55 °C for 30 s (annealing) and 72 °C for 60 s (extension), and finally maintaining 72 °C for 7 min. The amplification results were analyzed by electrophoresis onto a 1% agarose gel in TAE (Tris-acetate EDTA) buffer, the gel was run at 100 V for 60 min.

*2.15. Molecular Docking DNA/g-POCN*

The protein–ligand docking analyses were carried out using HEX 8.0 docking software. This software can superpose pairs of molecules using only their 3D shapes. It avails Spherical Polar Fourier (SPF) correlations to accelerate the calculations and has built-in graphics to visualize the docking effect. The parameters applied for the docking process via HEX docking were: Correlation type—Shape + Electro + DARS, FFT Mode—3D fast life, Post Processing-OPLS minimization, Grid Dimension—0.6, Receiver range—180, Ligand Range—180, Twist range—360, Distance Range—40, solutions—2000. The receptor consisted of a short dsDNA sequence downloaded from Protein Data Bank (5J3G). The ligand was modeled using ChemDraw, the ligand (g-POCN-modeled fragment) consisted of a six functionalized melem g-CN units with Phosphorus and Oxygen (Figure S3). Avogadro was used to minimize the ligand energy using the force field: MMFF94, then optimized in Chimera 1.14 software (adding hydrogens) and saved as a .pdb file. The lower obtained energies by HEX (KJ/mol) have been considered as spontaneous interactions, the HEX software scored the obtained poses based on Spherical Polar Fourier correlations, it could be considered as macromolecular–macromolecular rigid docking. High-resolution spherical polar docking correlations are performed over the resulting receptor surface patches [27].

The default settings used for the program to perform an initial Steric Scan at N = 18, followed by a Final Search at N = 25, using just the steric contribution to the docking energy. In this mode, about all but the top 30,000 orientations are discarded after the Steric Scan. Most conventional FFT docking algorithms have to use rather large grids (e.g., 0.6 Ångstrom cubes) because the grid must accommodate all possible translations of the ligand about a stationary receptor [27]. The low energy poses obtained by HEX between the modeled ligand (g-POCN fragment) and dsDNA were visualized in Pymol 2.4.1 software.

## 3. Results

### 3.1. PXRD Patterns

The synthesized doped carbon nitride products were characterized by XRD patterns as a function of the time of reaction. In the case of O-doped carbon nitride (g-OCN), after 24 h of reaction, XRD patterns showed two distinct peaks at 28.27° and 10.76° assigned to (002) and (100) diffraction (Figure 2). Typical carbon nitride exhibits the same diffractions [28] but at 27.4° and 13.1°, the observed displacement in g-OCN describes an interlayer distance decrease, which might be related to the increase in the intermolecular forces between O-doped layers and the diminished interlayer space. In general, the additive effect of the high oxygen-doped carbon nitride produces a remarkable increment in the (100) plane distance. In accordance with Kharlamov et al. [28] the increase in the (100) plane distance exhibits the effect conveyed by the presence of oxygen in the molecular skeleton and the existence of functional groups such as -OH attached to the tectons. In the case of the simultaneous phosphorus and oxygen doping (g-POCN), the XRD patterns showed two peaks at 27.8° and 10.8° (Figure 2). As shown, a slight increment of the interlayer distance of (002) plane is noticeable. This might be ascribed to the presence of phosphorus, which has a more

voluminous covalent radius than oxygen, nitrogen, or carbon; indeed, such increment expands the interlayer distance in opposition to enhanced van der Waals' forces.

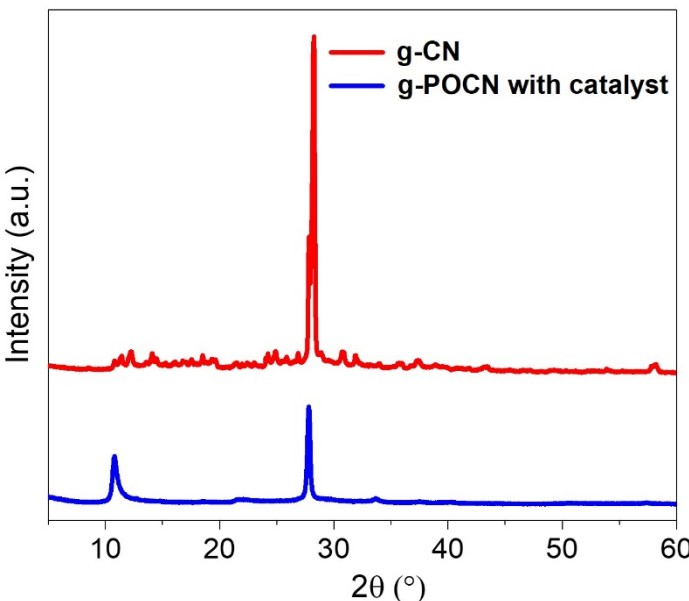

**Figure 2.** PXRD spectra of g-CN and Zn-catalyzed g-POCN.

*3.2. X-ray Photoelectron Spectroscopy*

XPS analysis of g-POCN (Figure 3a) revealed a chemical composition of 36.2% C, 43.2% N, and 20.6% O. Compared with quantitative elemental analysis (27.5% C, 42.5% N, 26.67% O, 3.26% H), the discrepancy of values is due to the type of analysis: XPS is merely a surface composition approximation, while quantitative elemental analysis is an absolute chemical determination. The calculated empirical formula was $C_{1.38}N_{1.83}H_{1.96}O$, and the content of oxygen is the highest reported in O-doped carbon nitride materials to date [29,30]. Hence, we propose that such a type of O-doped carbon nitride would be referred to as highly oxidized graphitic carbon nitride (g-HOCN).

The deconvoluted high-resolution spectra of C1s in g-POCN (Figure 3b) showed seven possible carbon bonds. These signals are attributed to C=H bond (283 eV); aromatic C-sp$^2$ (284.4 eV); C-N (285.7 eV) [31]; C-O (287 eV); C=O and COOH groups (288.5 eV) [32,33]; O=C-O group (289.9 eV); and the $\pi - \pi$* (HOMO-LUMO) transition (at 291.2 eV).

The deconvoluted high-resolution spectra of O1s in g-POCN (Figure 3c) showed six possible oxygen bonds. The signals at 528.9 and 530 eV probably correspond to the Zn-O bond produced by the oxidation of metallic Zn [34]. The peaks at 531.4 and 532.7 eV are in the typical energy range of the N-C-O and C-O/C=O groups, respectively [35,36], the peak at 533.8 eV is attributed to C-OH [37] and COOH group (534.7 eV) [38].

Figure 3d shows the deconvoluted high resolution spectra of N1s in g-POCN. The signals at 398.3 eV are due to the presence of pyridine N, i.e., an aromatic heterocycle (C=N-C) [31]; the signal at 399.7 eV is usually attributed to tertiary nitrogen N-(C)$_3$, ref. [39] which confirms the polycondensation of urea [40] although there may also be a pyridine N [41] contribution; the 401.4 eV peak is attributable to the N-H [42] bond; the bond at 403.8 eV may be attached to N-O [43]. The deconvoluted high-resolution spectrum of P2p in g-POCN (Figure 3e) showed five possible phosphorous bonds. The peaks at 131.1 and 131.5 have the typical P-C binding energy being 1 or 2 eV lower than for P-N [31], the signals at 133.1 and 134.3 eV are assigned to P-N/P=N [39] species; the peak at 135 eV may be associated with P=O bond [42].

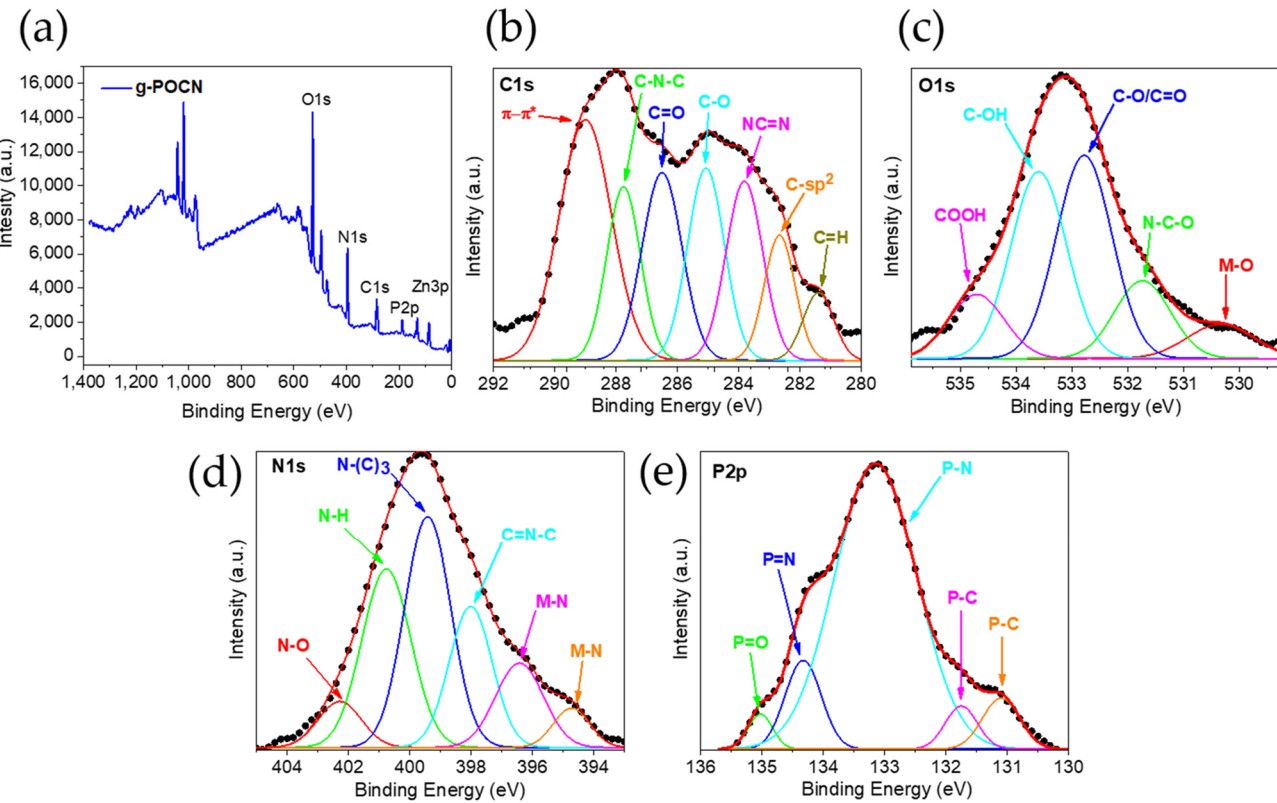

**Figure 3.** (**a**) XPS survey spectra of g-POCN, peak-fitted high-resolution (**b**) C1s, (**c**) O1s, (**d**) N1s, (**e**) P2p XPS spectra of g-POCN.

### 3.3. Fourier Transform Infrared Spectroscopy

FTIR spectrum of g-POCN (Figure 4) showed sharp and intense bands between the range of 3300–3450 cm$^{-1}$, corresponding to symmetric and asymmetric stretching of ($-NH_2$) groups [44]; vibrations at frequencies can be attributed to vibrational modes of intertectonic interactions between -N*H-. For instance, the band at 3200 cm$^{-1}$ may be attributed to the overtone of the scissors' vibration of -NH$_2$ [45,46]; meanwhile, the band at 3237 cm$^{-1}$ is possibly due to the stretching of the -OH group [47]. According to McMillan et al. [48] the broad bands between ~3100 and ~3030 cm$^{-1}$ are assigned to the formation of hydrogen bonds between -N*H- groups; however, the band extends up to about 2800 cm$^{-1}$, something unusual in other reported carbon nitrides.

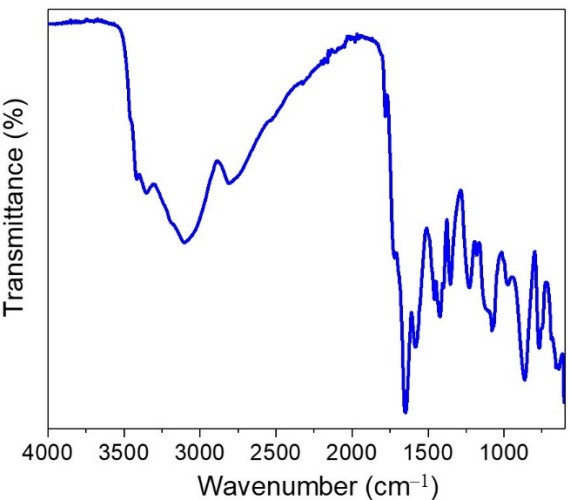

**Figure 4.** FTIR spectra of g-POCN.

The 1200–1650 cm$^{-1}$ region presented multiple absorption bands that, due to their intensity and position, are typical of g-CN-type structures, melem [2,6-triamino-s-heptazine $C_6N_7(NH_2)_3$], heptazine [$(C_6N_7)H_3$], melon $(C_6N_9H_3)$n, and other CN hetero rings [49]. The bands at 869 cm$^{-1}$ and 763 cm$^{-1}$ are particularly important since they are associated with vibrational modes out of the plane of the ring of s-triazine and its derivatives, a region considered as a fingerprint of these compounds [28,44,50].

The 1180 cm$^{-1}$ signal can be ascribed to the -C-O-C- stretch, oxygen that is suspected to be intra-cyclic [23]. The signals at 1089 and 1026 cm$^{-1}$ have the intensity attributed to the stretching of P=O, P-OR, or $PO_3/PO_4$ groups, respectively. The bands at 900 cm$^{-1}$ and 775 cm$^{-1}$ are associated with vibrational modes of the s-triazine ring and derivatives of cyanuric acid. The vibration at 900 cm$^{-1}$ is especially intense and relatively sharp, according to Seifer et al. [50], this could be assigned to the N-tert position of the cyanuric ring (in our case, a polycondensate cyamelurate derivative) coordinated with a metal atom.

### 3.4. Nuclear Magnetic Resonance

The $^{13}$C spectrum of g-POCN (Figure 5a) shows eight types of carbons. The high field signals at 25 and 67.5 ppm (not showed in the Figure) are attributed to ethanol as residual solvent. The 146.7 ppm signal correspond to an internal Cα-type ring. The carbons at 150, 150.4, and 151.5 ppm have shifts associated with a bay Cβ-type position, which chemical environment is influenced by electronegative groups and the presence of oxygen in the heterocycle [51–54]. The signals observed at high frequencies 156.7 and 158 ppm can also be ascribed to the Cβ bay carbon type.

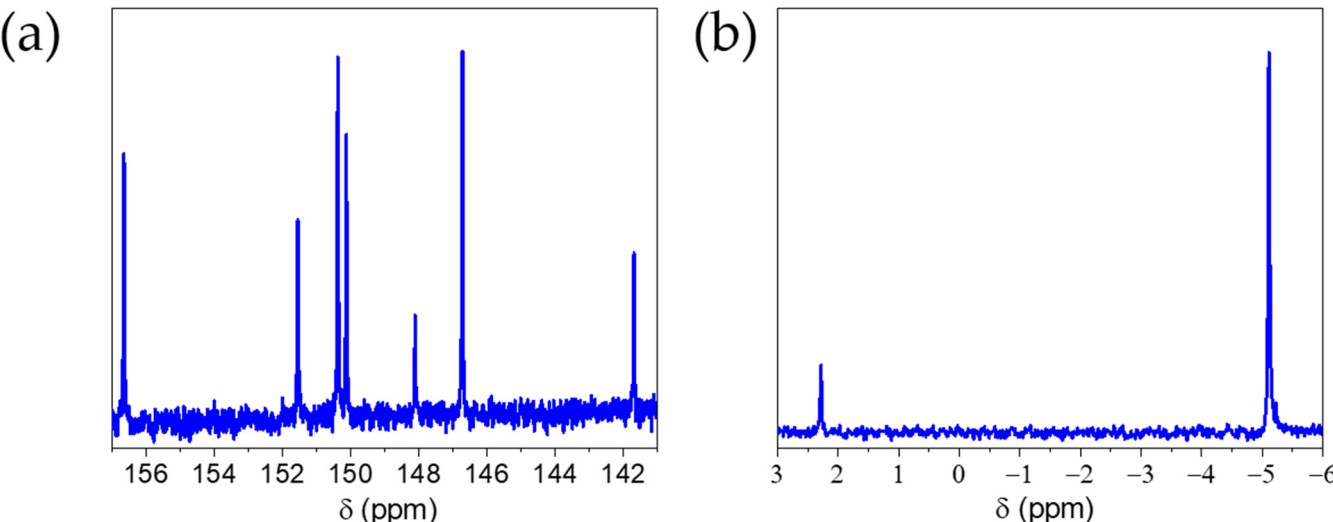

**Figure 5.** (**a**) $^{13}$C RMN spectra and (**b**) $^{31}$P RMN spectra of g-POCN.

The $^{31}$P spectrum of g-POCN (Figure 5b) shows two signals at -5 and +2.3 ppm, which, as discussed by similar investigations [39,55], suggest the presence of the terminal P=O in the ring or the $PO_4$ groups [42].

The signal at -5 ppm can be attributed to the intracyclic modification. Interestingly, previous reports describing P-doped materials have also found the signal at -5 ppm shifted to the range from −7 to −8 ppm, which was assigned to the N-P-N groups of the ring, or to a lesser extent to a possible intramolecular P-N substitution in terminal Cα position of the type [$NH_2$(N=P-N)OH]; such displacements are observed in phosphazenes with pentavalent phosphorus attached to three nitrogens and an electron donor group [56]. In the case of g-POCN, these observations seem to support the formation of tectons with P in the terminal Cβ position and possible phosphate groups attached to the ring; Zn also increases the phosphorus content in the g-OCN system.

### 3.5. Elemental Analysis

The calculated stoichiometry for the reaction product U+P Zn is $C_{1.28}N_{1.99}H_{3.065}O_{2.33}P_{0.46}$, which lowest integer formula can be generalized to $C_3N_4H_6O_5P$. The C/N ratio is 0.75, which can be classified as a carbon nitride with an ideal ratio [57]. On the other hand, the phosphorus content was increased with respect to the uncatalyzed reaction of Urea + $P_2O_5$ (Table 1), which indicates that the metallic Zn acts as a catalyst, promoting an increase in the phosphorus content in the structure.

**Table 1.** Elemental Analysis of Solvothermal Reaction Urea + $P_2O_5$ catalyzed by Zn.

|  | Catalyzed |
| --- | --- |
| **Element** | **Content** |
| Carbon | 15.44% |
| Nitrogen | 27.93% |
| Oxygen | 39.38% |
| Hydrogen | 3.065% |
| Phosphorus | 14.17% |

### 3.6. TEM

The TEM micrographs of g-POCN (Figure 6a) revealed micrometric sheets with a regular shape. Compared to the two-dimensional networks of g-OCN, this doped g-POCN presents a laminar morphology similar to some reported graphene materials [58,59]. The chemical modification of the structure of g-POCN leads to the formation of sheets of a lesser thickness and greater order with respect to g-OCN; an observation corroborated by the XRD analysis, which suggested the appearance of clusters with a smaller number of sheets. The sheets have a natural tendency to fold and roll, as seen in Figure 6b.

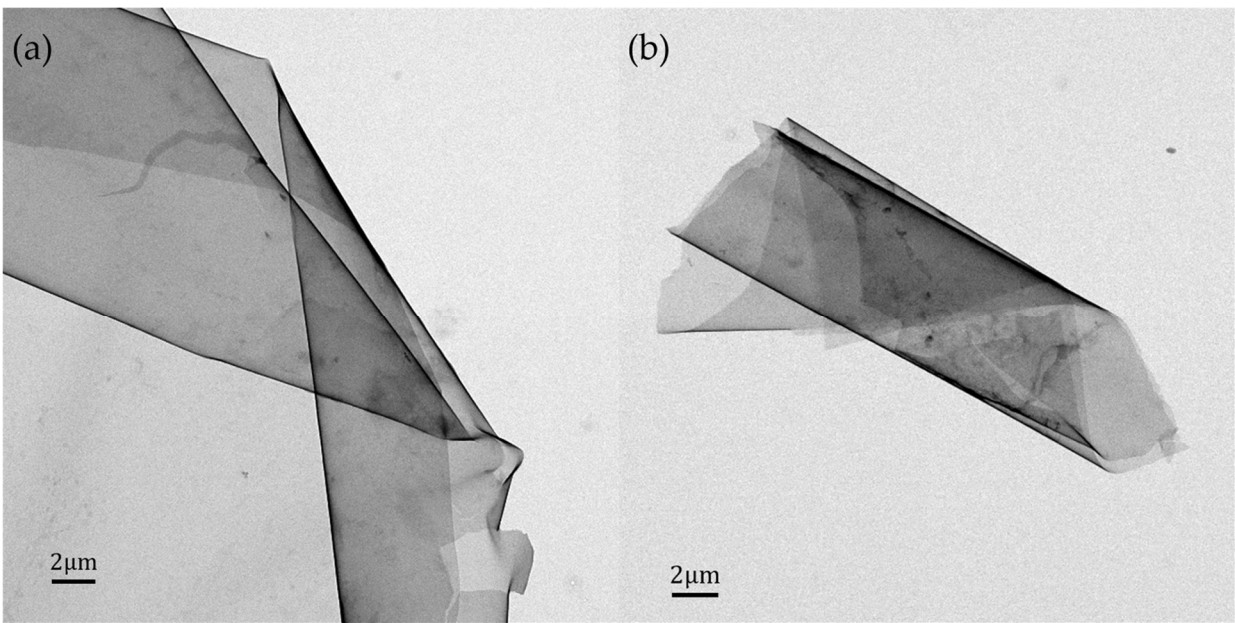

**Figure 6.** TEM micrographs of (**a**) g-POCN and (**b**) g-POCN sheet roll.

### 3.7. AFM

The AFM analysis of g-POCN (Figure 7a) confirmed the existence of discal sheets with large differences in size, from micrometric to submicrometric, and the presence of nuclei with greater stacking. The thickness calculated for a sheet at its thinnest edges was 1.29 nm (Figure 7b; the surface analyses are shown in Figure S2), which is a typical range for lamellar structures such as GO, g-$C_3N_4$, BCN, among others [60–62]. In general,

the topography of the image showed that most of the particles have a thickness close to approximately 9–10 nm.

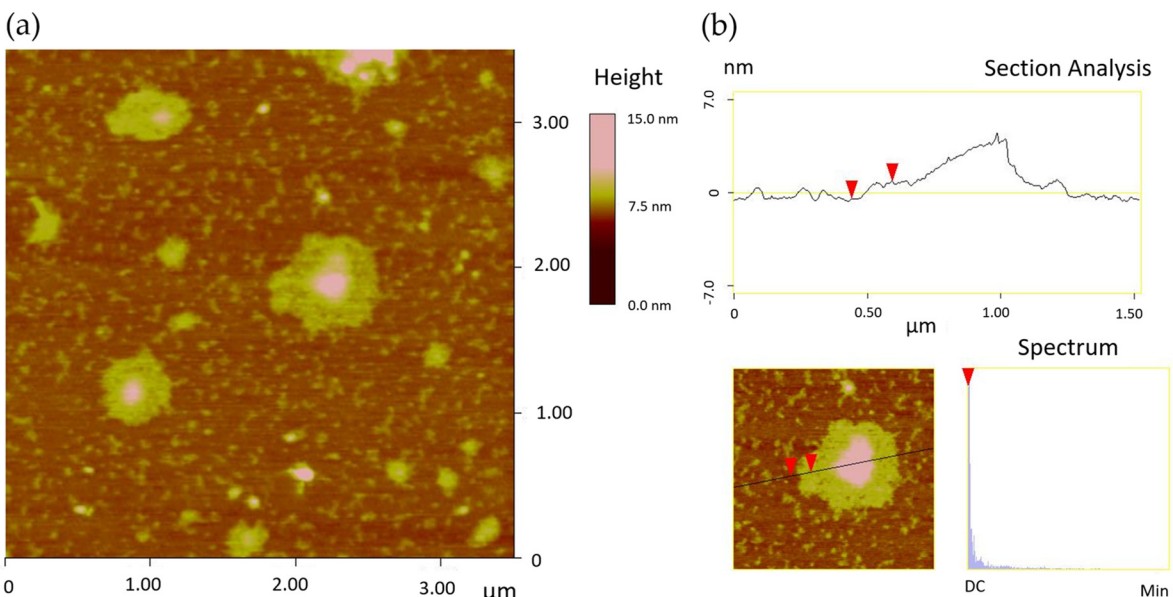

**Figure 7.** (**a**) AFM micrograph of g-POCN and (**b**) thickness analysis for a sheet of g-POCN.

*3.8. DNA Extraction*

The synthesized g-POCN was used for the extraction of genomic DNA (gDNA) from *P. argentatum*. The extraction of gDNA from a vegetal matrix was chosen because plant cells are usually complex matrices with low extraction yield through conventional methods. In this context, specialized extraction kits are the most common option to obtain gDNA with a high purity [63], however, they are expensive and with little freedom to modify the protocol on demand. *P. argentatum* is a desertic plant, which is endemic in north Mexican deserts; this species represents a commercial interest due to the high natural content of elastomers; our research group possesses certain expertise in the genetic characterization of this plant. However, it is extremely difficult to extract its DNA, as well as obtaining an acceptable purity, integrity, and yield using commercial methods; thus, we evaluated specifically this specimen here, in order to show the feasibility of the g-POCN DNA extraction method to isolate DNA from a complicated vegetal matrix. However, additional DNA extractions might be performed involving animal cells, bacteria, and fungi to demonstrate its universal use. On the other hand, the use of nanomaterials for gDNA extraction of animal cells and bacteria has been previously demonstrated, whose advantages are reflected in higher yields, shorter extraction times, and lower costs [64,65]. Table 2 shows the gDNA extraction yields using different methods. The DNA concentration for the extraction using the kit, isopropanol, and g-POCN with 50 μL and 30 μL was calculated at 97.4 ng/μL, 296.24 ng/μL, 365.6 ng/μL, and 316.35 ng/μL, respectively. Note that the highest gDNA yield was obtained with the addition of a larger amount of g-POCN, being almost four times more efficient than the kit. In addition, the purity of gDNA was calculated as the relationship between the absorptions (measured as optical density or OD) at 260/280 nm.

**Table 2.** gDNA extraction yields and relationship between the absorptions at 260/280 nm using different methods.

| Extraction Method | Total Concentration gDNA (ng/uL) | | OD (260 nm)/OD (280 nm) | |
|---|---|---|---|---|
| | Elution 1 | Elution 2 | Elution 1 | Elution 2 |
| g-POCN 30 μL | 316.35 | 142.15 | 1.88 | 1.9 |
| g-POCN 50 μL | 365.58 | 86.13 | 1.85 | 1.75 |
| Isopropanol | 296.24 | - | 0.7 | - |
| Qiagen Kit | 97.39 | - | 1.79 | - |

The OD 260/280 ratio was in the range of 1.85 to 1.88 for g-POCN and 1.79 for the kit. On the other hand, the common extraction method carried out with isopropanol, despite having an extraction yield close to g-POCN, has a 260/280 nm ratio equal to 0.7, which indicates a low purity in the extraction, a phenomenon related to the low selectivity of the precipitation of gDNA with this alcohol. Since the ideal purity is equal to a 260/280 nm ratio of 2.0, these results demonstrate the superior extraction capacity of g-POCN in comparison to the conventional method and the commercially available kit. Further analysis might be required to calculate the OD 260/280 ratio in order to elucidate if the samples are polysaccharide free; nevertheless, based on the above results, it is reasonable to believe that the extraction method does not present additional contaminations [66].

*3.9. DNA Extraction by Second Elution*

To examine the possible capacity of g-POCN to extract gDNA that was not completely eluted, we analyzed a second elution (E2) with TE on the recovered g-POCN. As shown in Table 2, the second elution allowed extraction yields between 142 and 83 ng/μL, depending on the amount of g-POCN. In addition, the obtained 260/280 nm ratios were between 1.75 and 1.9 (Table 2), indicating that, with respect to other methods, the extraction yield can increase almost five times by performing two elutions on the g-POCN. Thus, these observations allow us to propose g-POCN as a potential candidate to carry out highly efficient gDNA extractions.

Figure 8 shows the results of DNA electrophoresis in agarose gel of the different extraction methods. In all cases, defined stripes were obtained, which means that the extracted gDNA maintains a high integrity without fragmentation or degradation. Extraction with g-POCN (50 μL, lane 3) showed a retained band close to the well of the gel, which indicates the presence of molecules of very high molecular weight. This can be related to nanofragments of g-POCN that did not precipitate in the elution step, which retained part of the gDNA. In the case of extraction with g-POCN (30 μL, lane 5), this did not occur, and the area of the well was completely free; thus, demonstrating that a smaller amount of g-POCN is still equally efficient but allows a successful elution as there is no excess material in the medium. The second elution tests with g-POCN (lanes 4 and 6) demonstrated the feasible recovery of additional amounts of gDNA with optimal characteristics.

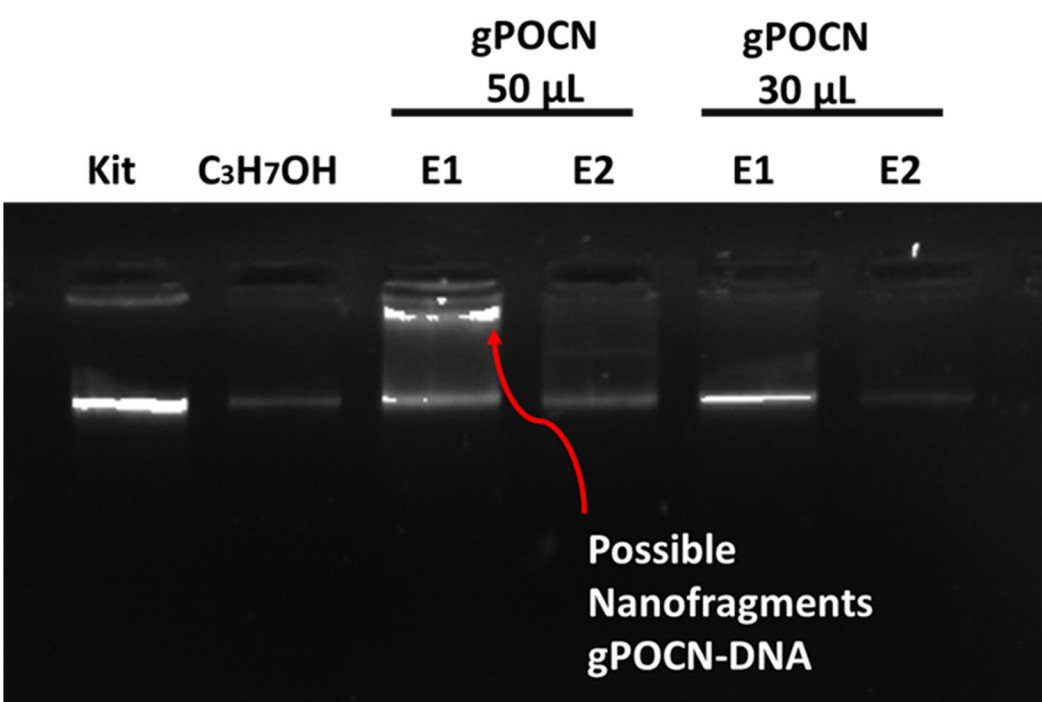

**Figure 8.** DNA extraction analyzed by electrophoresis onto a 1% agarose gel. Lane 1, extraction with the commercial Qiagen Kit; lane 2, extraction with the isopropanol method; lane 3, extraction with g-POCN (50 μL) first elution; lane 4, extraction with g-POCN- 50 (μL) second elution; lane 5, extraction with g-POCN (30 μL) first elution; lane 6 extraction with g-POCN (30 μL) second elution.

In general terms, this study shows that the extraction of gDNA from complex plant matrices with g-POCN is highly efficient, opening possibilities for future studies of RNA, plasmid, and DNA extraction from animal cells. Furthermore, in relation to the time required for liquid extraction methods, which usually take several hours or even days [10], g-POCN represents a potential candidate for simple, inexpensive, and fast DNA extractions.

*3.10. Molecular Docking DNA/g-POCN*

To further analyze the excellent capacity of g-POCN in the extraction of gDNA, molecular docking was used in order to enlighten the interactions between g-POCN and gDNA. Molecular docking examines processes in which two molecules fit together in a three-dimensional space. This represents a useful tool in structural molecular biology and drug design that provides valuable information on the specific interactive trends of related molecules and predicts the most favorable modes of interaction between a receptor and a ligand [67]. In this study, a small DNA sequence was defined as a receptor and a modeled g-POCN fragment as a ligand, according to the most probable chemical functionalities determined by the various spectroscopy techniques.

Figure 9a shows the most favorable pose determined in the HEX software. The interaction with the lowest energy is I ($-453.6$ KJ/mol), which means that this interaction is thermodynamically advantageous with respect to others. As seen, pose I shows an intercalation of a single nitride layer with respect to the DNA in the minor groove. These results suggest that, in experimental conditions, there are strong intercalation interactions between the edges of the nitride layers and the different overlapping of long DNA chains. Thus, we could explain the high affinity of DNA for g-POCN observed in our earlier experimental results. In this regard, previous studies have shown that carbon nitride nanolamines (g-$C_3N_4$) have a higher affinity for ssDNA with respect to dsDNA; a feature used as an advantage for the development of g-$C_3N_4$/ssDNA biosensors [23]. However, such affinity is because g-$C_3N_4$ has high hydrophobicity, and in this sense, the exposed nitrogenous bases of the ssDNA usually interact through π-stacking with the g-$C_3N$ [24,68].

In the case of g-POCN, having a modified carbon nitride structure with a high content of oxygen and phosphorus, the hydrophilicity of the material increases considerably.

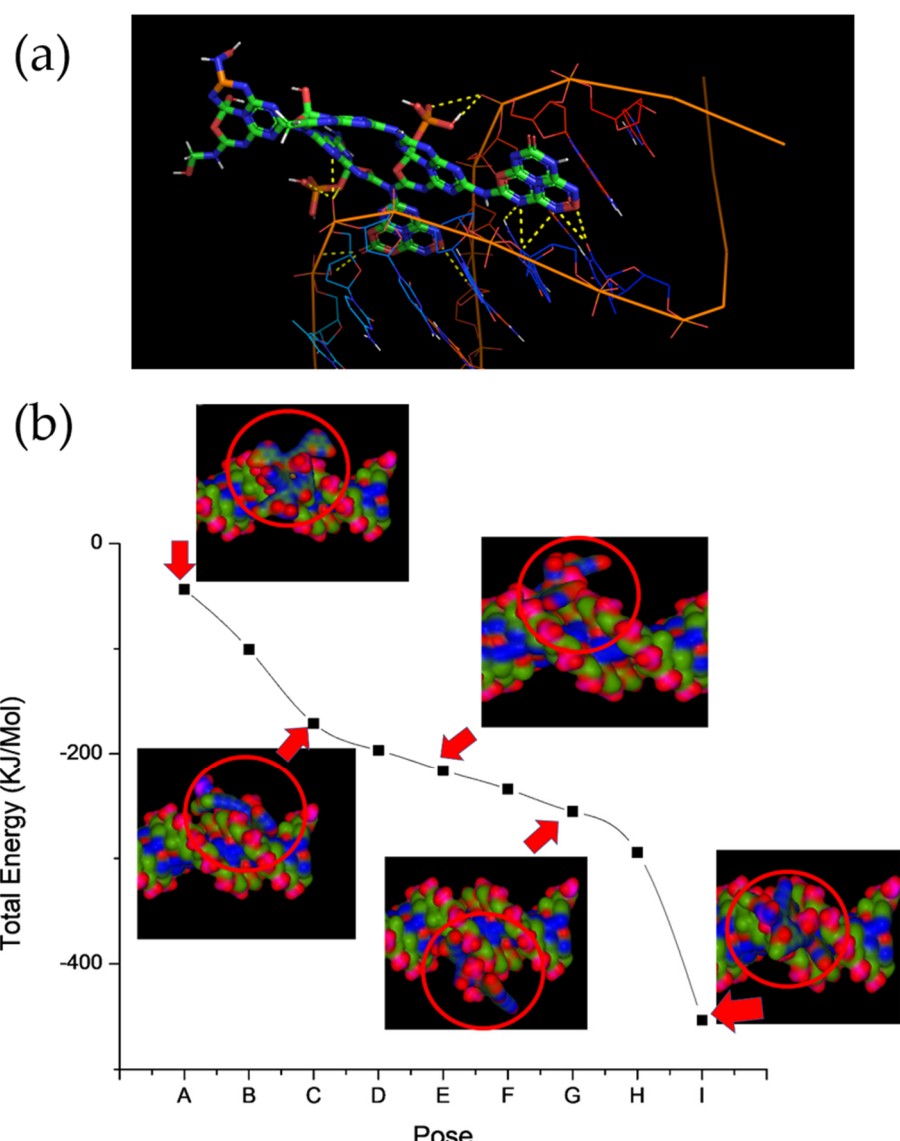

**Figure 9.** (**a**) Molecular Docking between a small DNA sequence, defined as a receptor, and a modeled g-POCN fragment, defined as a ligand (the most stable pose). (**b**) The energies calculated for different poses between the g-POCN fragment and the dsDNA.

According to Figure 9a, the van der Waals contributions can be seen as well as hydrogen bonds between the different chemical functionalities of g-POCN with the nitrogenous bases, deoxyribose, and phosphates of the dsDNA skeleton. In Figure 9b, the energies calculated for perpendicular interactions between the g-POCN, and dsDNA (pose A) are less favorable in the order of -40 KJ/mol. This is in agreement with reported theoretical calculations for interactions between graphene oxide and dsDNA [69], indicating that the interactions of the functional groups between both molecules are limited by the orientation, where apparently the hydrogen bonds diminish the intercalation of the edges of the sheets with the major groove. In this manner, the dsDNA begins to decrease the energy of the poses until reaching the most energetically favorable interaction when intercalation occurs with the minor groove of dsDNA.

It is also worth mentioning that interactions can also occur due to π-stacking, van der Waals, and hydrogen bonding between the nitrogenous bases of the DNA chains and the

surface of the g-POCN. The interaction due to π-stacking in dsDNA has been theoretically described by molecular dynamics between dsDNA and GO; it consists of a dehybridization of the terminal segments of the double helix, exposing some nitrogenous bases that are associated with the surface of the material through the mentioned interaction [70]. Therefore, we could put forward that these types of interactions are taking place in an analogy between dsDNA and g-POCN. Together, it seems that intercalation, perpendicular interaction, and interaction of the terminal segments of DNA are the three possible ways in which DNA interacts with g-POCN. These results further supported the experimentally found high affinity between DNA and g-POCN and are the basis of the high extraction efficiency. Remarkably, although the extraction efficiency of dsDNA with g-POCN is considerably better than other extraction methods, it is low enough to be able to elute DNA in the presence of TE. Otherwise, larger and more favored interactions on a larger scale would deliver the observed phenomenon between g-$C_3N_4$ and ssDNA, where the elution of DNA is highly complex [68]. Thus, g-POCN exposes an advantageous and optimal degree of functionality to adsorb as much dsDNA as possible without compromising the ease of subsequent elution.

### 3.11. DNA Amplification by PCR

To verify the integrity, consistency, and absence of interfering complexes as a result of the DNA extraction processes, the extracted gDNA was analyzed through PCR amplification. A fragment of the ITS1–ITS2 region in *P. argentatum* was amplified, generating a fragment of approximately 700 bp; the selection of this ITS rDNA fragment to be amplified obeys fundamentally the suitability and standardized conditions previously obtained for *P. argentatum* in our laboratory. However, the amplification of single copy genes remains as a necessary future experiment to test the robustness of the g-POCN DNA extraction method. Figure 10 shows the agarose gel electrophoresis of the PCR products obtained from the different extraction methods used in this investigation. The extraction method using the Qiagen Kit and Ispropanol are positive (lanes 4–5 and 8–9), which is in accordance with the convention of such techniques in gDNA extraction from plant matter. Similarly, the extraction by g-POCN (lanes 6–7) offers a successful amplification of the selected region for *P. argentatum*, reliably demonstrating the robustness of the extraction method with g-POCN to obtain gDNA of sufficient quality to carry out its use in PCR amplification. Two concentrations of gDNA were studied for amplification: 0.25 and 0.02 µL, the bands of the amplified region are more saturated in the case of the higher concentration of gDNA. In the case of the lower concentration, the amplification was successful in all the extractive methods. This is consistent in all the extraction methods as expected, meaning that g-POCN is a comparatively equivalent method to the commercial kit and isopropanol extraction and a potential candidate for situations where there are only small amounts of DNA to be extracted. Likewise, these observations confirm that g-POCN does not interfere in any way in the PCR amplification, which by itself is ideal since most of the reagents used in liquid–liquid extraction have a certain degree of inhibition in this process [10]. In the same way as the use of solid columns, as in the case of commercial kits, or the use of nanomaterials such as GO or magnetic nanoparticles have shown no interference in the polymerase reaction [9,18], g-POCN does not present mechanisms of polymerase chain-reaction inhibition.

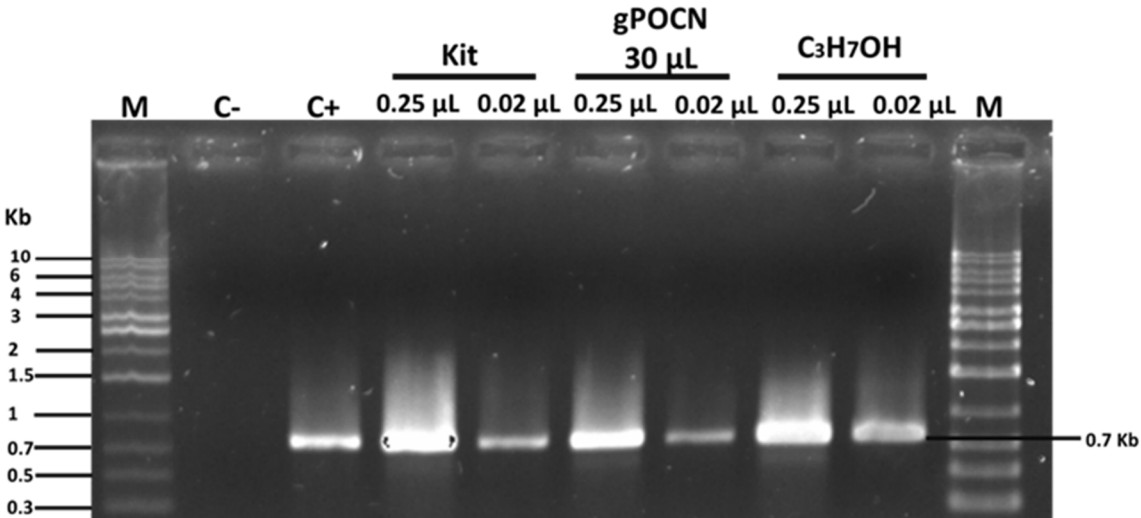

**Figure 10.** Analysis of PCR products amplified with ITS primers using different genomic DNA extraction methods. Lane 1 and 10, molecular weight markers; lane 2, negative control; lane 3, positive control; lanes 4 and 5, DNA extracted by a commercial kit using different concentrations; lanes 6 and 7, DNA extracted by g-POCN using different concentrations; lanes 8 and 9, DNA extracted by isopropanol using different concentrations.

## 4. Conclusions

Doped carbon nitride structures were successfully synthesized through a one-pot solvothermal approach using powder zinc as the catalyst and urea and phosphorus pentoxide as the carbon, nitrogen, and phosphorus source. Extraction of gDNA with g-POCN was evaluated by comparing this method with commercial and conventional methods. The yields as well as the integrity and quality of the extracted gDNA by g-POCN are comparable to a commercial extraction kit and isopropanol extraction, and no degradation of any kind was observed. Molecular docking revealed that the high extraction efficiency of g-POCN is related to the planarity of the material, nanometric thickness of the sheets, and the great variety of groups with the presence of oxygen, nitrogen, and phosphorus, which together generate electrostatic interactions, van der Waals, and hydrogen bridges as well as intercalations between the double helix that efficiently adsorb dsDNA. Furthermore, under suitable elution conditions, it is possible to easily remove a high concentration of gDNA from the material. Thus, experimental and molecular modeling results showed that g-POCN could be used as a low-cost, simple, and fast extraction method, without interference in PCR amplification, scalable and optimal in situations where only small amounts of gDNA can be extracted. We conclude that g-POCN represents a promising DNA extraction platform that could further stimulate research on doped carbon nitride materials applied to the extraction of RNA, plasmids, ssDNA, in broad DNA sources.

**Supplementary Materials:** The following supporting information can be downloaded at: https://www.mdpi.com/article/10.3390/c8040068/s1, Figure S1: Elemental Analysis Report Methods; Figure S2: AFM data obtained from the surface analysis; Figure S3: g-POCN modeled fragment in ChemDraw, optimized by Avogadro and Chimera software.

**Author Contributions:** Conceptualization, M.E.M.-C., J.B.-M. and J.R.-G.; methodology, T.C., J.R.-G. and A.B.-V.; investigation, M.E.M.-C.; analysis and characterization, A.B.-V., T.C. and J.R.T.-L.; resources, M.E.M.-C., S.F.-T. and A.M.R.-H.; writing-original draft preparation, M.E.M.-C. and A.M.R.-H.; writing-review and editing, R.D.-d.-L., V.D.L.-I. and I.M.; visualization, M.E.M.-C., R.D.-d.-L. and A.M.R.-H.; project administration, M.E.M.-C. and S.F.-T.; funding acquisition, M.E.M.-C., A.M.R.-H. and R.D.-d.-L. All authors have read and agreed to the published version of the manuscript.

**Funding:** The authors acknowledge the financial support of the Fund for Scientific and Technological Research (FONCyT) through the Project COAH-2020-C14-B007: Biomimetic synthesis of bioinspired

semiconductor conjugated copolymers as neuronal interfaces, and the Mexican National Council of Science and Technology (CONACYT) through the Basic Science Project CB 2016-287954-Q: Hematin supported on nanostructured carbon nitride layers: Biomimetic catalyst for the synthesis of polyaniline. We also acknowledge the financial support of the Research Center in Applied Chemistry (CIQA) through internal projects 6674 and 6677.

**Institutional Review Board Statement:** Not applicable.

**Informed Consent Statement:** Not applicable.

**Data Availability Statement:** Not applicable.

**Acknowledgments:** In honor and loving memory of Jorge Romero, who has been a special part of this work and holds a special place in our hearts. The authors thank Dra. Yolanda Ortega-Ortega, José Díaz Elizondo, Martha Roa, Fabiola Castellanos, and Ricardo Mendoza for their technical support in the characterization of samples.

**Conflicts of Interest:** The authors declare no conflict of interest.

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
