# Peer review of "Easy Synthesis of Doped Graphitic Carbon Nitride Nanosheets as New Material for Enhanced DNA Extraction from Vegetal Tissues Using a Simple and Fast Protocol"

_carbon, 2022_

Round 1

Reviewer 1 Report

Manuscript carbon-2025272 entiteled "One-pot solvothermal synthesis of doped graphitic carbon nitride as biocompatible components for enhanced DNA extraction from vegetal matrices" and authored by Manuel Eduardo Martínez-Cartagena , Juan Bernal-Martinez , Arnulfo Banda-Villanueva , Víctor D. Lechuga-Islas , Teresa Córdova , Ilse Magaña , Roman Torres-Lubian , Salvador Fernández-Tavizón , Ana Rodríguez-Hernández and Ramón Díaz-de-León targets a hot topic that is potentially of high interest to the journal readers and to the scientific community as a whole. Despite the satisfactory design of the experiments and the valuable data presented several points needs unfortunately the attention of the authors before acceptance for publication of the manuscript:

1. The title of the manscript have to be changed the authors should point the importance and impact of their findings rathar than using general words that do not encourage the reader to go inside the manuscript. This point is critical for the citation potential of the paper.

2. Introduction section : a critical aspect of the study have not been discussed at all : the toxicity of the different protocols. The toxicity of the three different approaches have to be evaluated and discussed.

3. Results section : having spent many years extracting DNA from plant material the most critical aspect is the presence of polysaccharides that interfer with downstream applications. The authors have to justify this point ! have you evaluated OD (260 nm)/OD (280 nm)? please comment on this point it is critical for the readers !

4. Results section : have the authors tested another source for genomic DNA : another palnts to see if the protocol performs well across plant species ! I am not familiar with P. argentatum. Is it easy to extract DNA from this species ? Why you selected this palnt species? please enrich your manuscript your findings are nice and enriching discussions will give more merit in favour of your manuscript.

5. Results section : Have you tried your protocol to extract DNA from animal tissues or fungi ? please if so document this part.

6. Results section : please document why you amplified ITS rDNA ? why not to add other markers such as single copy genes that are more difficult to amplify than ITS rDNA which is relatively easy to amplify.

Finally I am really waiting to read an improved version of this manuscript that addressed vall these issues and that reach the journal standards for the wide interest of the journal readers and the scientific community. I will be happy to recommend this manuscript for publication once all these points addressed.

Best regards

Reviewer 2 Report

This manuscript is relevant to a new method for synthesizing doped CN structures for improved DNA extraction. This work includes novel features and is well written. However, the authors should perform various modifications to their manuscript:

In the Introduction section, the authors should present some more details about "new synthesis pathways" mentioned in the first paragraph, as well as solvothermal synthesis.

The authors should mention the number of each reference directly after "et al." (for example, "Hashemi et al. [6]").

More details should be presented regarding the procedure of DNA extraction with doped g-CN and at least 10 more references regarding similar studies in the relevant literature should be added and discussed in the Introduction section, in order to prove the novelty of the proposed method.

In section 2, a Table including the details about the materials used in this work should be added, as well as a flowchart indicating the experimental and analysis procedure of this research.

In line 249 "43,2% N" should be corrected to "43.2% N".

In line 254, a reference should be added regarding the statement  "the content of oxygen is the highest reported in O-doped carbon nitride materials to 253 date".

In Table 1, "Hidrogen" should be corrected to "Hydrogen".

More details should be added about the simulations conducted regarding Molecular Docking.

In page 13, the text should be split into at least 2 separate paragraphs. 

Round 2

Reviewer 1 Report

Dear authors,

I can now recommend your manuscript for acceptance.

Best regards

Reviewer 2 Report

The authors have performed all the necessary modifications to their manuscript. Thus, it can now be accepted for publication.